

# Exercise type, training load, velocity loss threshold, and sets affect the relationship between lifting velocity and perceived repetitions in reserve in strength-trained individuals

Gøran Paulsen[1,2], Roger Myrholt[1], Fredrik Mentzoni[2] and Paul Andre Solberg[2]

[1] Department of Physical Performance, Norwegian School of Sport Sciences, Oslo, Norway
[2] Norwegian Olympic and Paralympic Committee and Confederation of Sports, Oslo, Norway

## ABSTRACT

**Purpose.** To explore the relationship between bar velocity and perceived repetitions in reserve (pRIR) for the bench press and the squat exercises during multiple training sessions in strength-trained individuals.

**Methods.** Nineteen well-trained individuals (9♀ and 10♂, 26 ± 4 yr, 174 ± 8 cm, 74 ± 9 kg (mean ± standard deviation)) trained squats and bench press for six weeks. Within each week, they conducted three sessions with different loads, corresponding to ~77–79%, ~82–84%, and ~87–89% of one repetition maximum (1RM). The mean velocity was measured at the bar for all lifts, and the participants terminated each set based on a pre-set velocity loss threshold (20–60%). After every set termination, the participants reported pRIR.

**Results.** Based on 2,972 unique measurements, we observed trivial to very large individual correlations between the objectively measured mean velocity and the pRIR (average $r^2 = 0.3$ for both squat and bench press). Type of exercise (squat or bench press), velocity loss threshold, load, and sets affected the pRIR for a given mean velocity. Sex (females *vs.* males) and training weeks were unrelated to pRIR.

**Discussion and conclusion**. Our findings indicate that mean bar velocity and pRIR offer complementary—but not interchangeable—perspectives on strength training performance. Because pRIR was systematically influenced by exercise type, external load, proximity to failure, and set number, practitioners and researchers should interpret pRIR with caution and in the context of these variables.

Corresponding author
Gøran Paulsen, goranp@nih.no

# INTRODUCTION

Velocity-based strength training (VBST) has become a popular objective approach for managing exercise intensity/load and neuromuscular fatigue (*Bastos, Machado & Teixeira, 2024*; *Jovanovic & Flanagan, 2014*; *Maroto-Izquierdo et al., 2025*; *Suchomel et al., 2021*). Velocity-based strength training differs from traditional strength training by focusing on lifting velocity rather than loads expressed as a percentage of one repetition maximum

(1RM) or a specific number of repetitions. Typically, VBST is based on the mean velocity of the bar (ascending phase), measured by devices such as a linear encoder or an accelerometer (*Jovanovic & Flanagan, 2014*; *Weakley et al., 2021*). A prerequisite with VBST—and using mean velocity as a variable—is maximal effort in the ascending phase of each repetition (*Jovanovic & Flanagan, 2014*).

It is well established that desired strength training outcomes rely on load intensity (*e.g.*, % of 1RM) and volume (*Atha, 1981*). However, the effort in each lift (intention to move) and proximity to set failure can also be used to tune the adaptations to strength and power training (*Grgic et al., 2022*; *Kawamori & Newton, 2006*). While hypertrophy appears to be most effectively achieved with (near) failure sets, non-failure sets are recommended as preferable for developing power and maximal strength in well-trained individuals and athletes (*Carroll et al., 2018*). Managing and controlling neuromuscular fatigue during exercise may improve training quality and promote specific adaptations (*Izquierdo et al., 2006*; *Larsen, Kristiansen & van den Tillaar, 2021*; *Suchomel et al., 2021*).

As the mean velocity gradually declines with successive repetitions, a percentage loss of velocity or pre-set velocity threshold (in ms$^{-1}$) can be used to decide when to terminate the set. In practice, low velocity loss thresholds (*i.e.,* <25%) appear preferred for developing muscle power and maximal strength, while hypertrophy seems to require higher velocity loss thresholds (>25%), *i.e.,* close to contraction failure or RM (*Baena-Marín et al., 2022*; *Hickmott et al., 2022*; *Weakley et al., 2021*).

The mean velocity (ms$^{-1}$) has been demonstrated to correlate with both the percentage of 1RM and repetitions in reserve (RIR) (*Pelland et al., 2022*). Repetitions in reserve refer to the difference between the number of repetitions performed and the maximum possible repetitions in a set. However, it is evident that the relationship between mean velocity and RIR depends on the type of exercise, *e.g.*, the bench press and the squat, and the high inter-individual variability seems to necessitate individual adjustments for practical application (*Garcia-Ramos et al., 2018*; *Jukic et al., 2024*; *Moran-Navarro et al., 2019*).

An alternative to objective measures of mean velocity is the subjective evaluation of effort and fatigue through the rate of perceived exertion (RPE) and perceived repetitions in reserve (pRIR) (*Helms et al., 2016*; *Helms et al., 2020*; *Zourdos et al., 2016*). The main argument for using RPE and pRIR is autoregulation, which is an individual, intuitive, and dynamic way to account for biological day-to-day variations and readiness to train (*Suchomel et al., 2021*). Perceived RIR is undeniably simplistic, practical, and low-cost, with no need for technological devices.

Both the objective mean velocity (*Garcia-Ramos et al., 2018*; *Moran-Navarro et al., 2019*) and subjective pRIR (*Hickmott, Butcher & Chilibeck, 2024*; *Varela-Olalla et al., 2019*) have been reported to predict the actual RIR in a set with acceptable accuracy. The pRIR seems, however, more influenced by the proximity to failure and training status than the objective, mean velocity measurements (*Halperin et al., 2022*; *Moran-Navarro et al., 2019*). That said, the applicability of the mean velocity method is also debatable, as some find good accuracy and reliability (*Jukic et al., 2024*), while others concluded that the mean velocity does not appropriately align with actual RIR values across loads and sets (*Mansfield et al., 2023*).

Previous studies examining the relationship between mean velocity, RIR, and pRIR have done so in controlled laboratory settings, using only a few test sessions (*Garcia-Ramos et al., 2018*; *Jukic et al., 2024*; *Mansfield et al., 2020*; *Varela-Olalla et al., 2019*). In such experimental settings, when multiple conditions tested in series, *e.g.*, different loads and repetition ranges, the order and proximity of the conditions may affect the perceived efforts and ratings. First, the number of lifts and sets becomes restricted due to the development of neuromuscular fatigue, meaning that the total number of data points from each individual becomes limited. Second, the participants are, in fact, aware of the purpose of the study and may develop biased responses when assessing similarities or differences between the various conditions, compared to real-world scenarios where the focus is on the training itself (*i.e.,* the Hawthorne effect). This raises questions about the ecological validity of the available literature on the topic. An alternative approach is to collect mean velocity and pRIR data from consecutive real-life training sessions that vary in load intensity and proximity to failure.

This study explored how exercise type (bench press and squat), load intensity (% of 1RM), velocity loss thresholds, number of sets, and sex may influence the relationship between objectively measured bar mean velocity and pRIR across multiple strength training sessions in well-trained individuals.

We recognize both VBST and pRIR as potentially valuable tools for managing exercise effort and training specificity (*Helms et al., 2016*; *Weakley et al., 2021*), and our results may offer clearer insights into how these tools can be applied by individuals involved in strength training.

## METHODS

### Participants

Nineteen well-trained young adults participated in this study (Table 1). Participants were classified as "advanced" to "highly advanced" resistance-trained individuals according to *Santos et al. (2021)*. One male trained only the squat exercise, and one male trained only the bench press exercise due to shoulder and knee pains, respectively.

The inclusion criteria required volunteers to be young adults (<40 years of age), have at least one year of consistent bench press and squat practice—including the past year, and demonstrate a minimum relative 1RM (bar mass/body mass (kg/kg)) of 0.6 (females) or 1.0 (males) in the bench press and 1.0 (females) or 1.2 (males) in the squat.

After receiving written and oral information, all participants volunteered to participate in the study by signing an informed consent form. The study was approved by the local ethical committee at the Norwegian School of Sport Sciences (103-290819).

### Study design

The present study was based on data from a 6-week strength training period (18 sessions). The participants trained the bench press and the squat exercises in the same session. The mean velocity was tracked for all lifts. For the purpose of this study, we analyzed only the velocity of the last repetition in each set, along with the pRIR, which was recorded after
**Table 1 Baseline data.**

| | Females (*n* = 9) | | | Males (*n* = 10) | | |
|---|---|---|---|---|---|---|
| | Mean ± SD | | | Mean ± SD | | |
| Age (years) | 26 | ± | 5 | 26 | ± | 3 |
| Height (cm) | 169 | ± | 6 | 178 | ± | 6 |
| Body weight (kg) | 67 | ± | 7 | 81 | ± | 5 |
| Squat 1RM (kg) | 86 | ± | 18 | 141 | ± | 16 |
| Bench press 1RM (kg) | 51 | ± | 7 | 114 | ± | 11 |

**Notes.**

1RM, 1 repetition maximum; SD, standard deviation.

each set. The velocity loss threshold was calculated from the pre-planned velocities (see below).

After a 1RM test in bench press and squat, the participants were allocated to one of two groups. One group trained with a "low" velocity loss threshold (LVL), while the other group trained with a "high" velocity loss threshold (HVL). The training adaptations, such as changes in 1RM and local hypertrophy, to the LVL and HVL interventions are published elsewhere (*Myrholt et al., 2023*).

Of the 19 participants, four completed both the LVL and the HVL program; hence, seven males and four females completed the LVL program, and seven males and five females completed the HVL program. We used a linear mixed-effects model (LMM) that allows for individual differences in the number of data points, ensuring that each participant's contribution is appropriately weighted in the analysis. Note that this study includes three more participants than those analyzed in *Myrholt et al. (2023)*. The raw data is available for download.

## Rationale for the velocity thresholds and training volume

The LVL group trained with 20% and 30% mean velocity loss thresholds in squat and bench press, respectively (Table 2). The HVL group trained with 40% and 60% velocity loss thresholds in squat and bench press, respectively. These thresholds, which determined set termination once the velocity dropped below the specified percentage, were chosen based on previous studies (*Pareja-Blanco et al., 2017*; *Pareja-Blanco et al., 2020*; *Sanchez-Medina & González-Badillo, 2011*) and extensive pilot testing to ensure that the number of repetitions with the LVL program resulted in about half the number of repetitions per set as the HVL program (Table 2).

The training volume (total number of reps) between groups was matched, as the HVL trained with ∼3 sets and the LVL trained with ∼6 sets per exercise (Table 2). To achieve this, the HVL group was always three sessions ahead of the LVL group so that the LVL group's training volume could be adjusted weekly. Hence, we adjusted the number of sets per session for the LVL group to achieve approximately the same total number of repetitions as the HVL group.

**Table 2 Descriptive data of the training sessions.** Data are presented for the bench press and squat. The columns represent the three sessions conducted with different loads defined by bar velocity. The rows represent the two groups that trained with either low- or high-velocity-loss thresholds (LVL, bench press 30% and squat 20%; HVL, bench press 60% and squat 40%).

| | Bench press | | | Squat | | |
|---|---|---|---|---|---|---|
| | Low load $0.53$ ms$^{-1}$ $\sim$77% of 1RM | Moderate load $0.45$ ms$^{-1}$ $\sim$82% of 1RM | Heavy load $0.38$ ms$^{-1}$ $\sim$87% of 1RM | Low load $0.70$ ms$^{-1}$ $\sim$79% of 1RM | Moderate load $0.60$ ms$^{-1}$ $\sim$84% of 1RM | Heavy load $0.49$ ms$^{-1}$ $\sim$89% of 1RM |
| Number of reps per set | | | | | | |
| LVL | 4.7 ± 1.6 | 3.9 ± 1.5 | 2.7 ± 1.2 | 4.2 ± 2.0 | 3.2 ± 1.5 | 2.1 ± 1.2 |
| HVL | 9.5 ± 3.2 | 7.9 ± 3.1 | 6.0 ± 2.4 | 9.5 ± 4.2 | 7.0 ± 3.2 | 5.1 ± 2.4 |
| Number of sets per session | | | | | | |
| LVL | 6.0 ± 1.1 | 5.3 ± 1.1 | 5.3 ± 1.6 | 5.9 ± 2.4 | 5.4 ± 1.8 | 5.5 ± 1.8 |
| HVL | 3.0 ± 0.2 | 3.1 ± 0.3 | 2.9 ± 0.4 | 2.9 ± 0.3 | 3.0 ± 0.3 | 3.0 ± 0.3 |
| pRIR | | | | | | |
| LVL | 2.6 ± 1.0 | 2.2 ± 1.0 | 1.5 ± 1.0 | 2.6 ± 1.0 | 2.0 ± 1.0 | 1.4 ± 0.9 |
| HVL | 1.6 ± 0.9 | 1.3 ± 0.9 | 1.2 ± 0.9 | 2.2 ± 1.0 | 1.9 ± 1.0 | 1.2 ± 1.0 |
| Velocity loss | | | | | | |
| LVL | 31.3 ± 7.4 | 31.8 ± 8.5 | 31.2 ± 10.9 | 21.2 ± 5.2 | 22.0 ± 6.3 | 21.3 ± 8.0 |
| HVL | 61.5 ± 7.5 | 59.5 ± 8.4 | 55.8 ± 10.7 | 41.3 ± 6.5 | 40.3 ± 6.0 | 39.6 ± 8.2 |
| Velocity (ms$^{-1}$) | | | | | | |
| LVL | 0.36 ± 0.04 | 0.31 ± 0.04 | 0.27 ± 0.04 | 0.55 ± 0.04 | 0.48 ± 0.04 | 0.40 ± 0.04 |
| HVL | 0.20 ± 0.04 | 0.18 ± 0.04 | 0.17 ± 0.04 | 0.40 ± 0.05 | 0.36 ± 0.04 | 0.30 ± 0.04 |

**Notes.**

1RM, 1 repetition maximum; HVL, high-velocity-loss threshold group; LVL, low-velocity-loss threshold group; pRIR, perceived repetitions in reserve.

### Measuring mean velocity of the bar

A linear encoder tracked the duration (time, in seconds) and displacement (in meters) of the bar in all lifts (Musclelab, Ergotest; 200-Hz sampling rate and 0.019-mm resolution). The encoder was attached to the powerlifting bar on the right-hand side (10 cm outside the knurl mark). Care was taken to position the encoder to measure the vertical bar displacement. This was inspected in all sessions, and individual marks (tape) on the floor were used to ensure similar foot positioning from set to set in the squat. Similarly, the shoulder positioning and the placement of the encoder were individually adjusted in the bench press. A slight horizontal bar movement in the bench press is unavoidable, but care was taken to ensure the setup was similar for all sets at the individual level. All velocities are given as the mean velocity of the lifting (ascending) phase (calculated by Musclelab's proprietary software Ergotest, version 10.200.90.5097). The braking (descending) phase was not included in this study. Compared to 3D kinematic analyses of free-weight squats (Qualisys, Gothenburg, Sweden; sampled at 300 Hz), the coefficient of variation (CV) for mean velocity measurements is 1.9% (unpublished data from our laboratory).

### Perceived repetitions in reserve

Prior to the training period, the participants were informed about the concept of RIR and pRIR. They were instructed to verbally express their pRIR immediately after each set. The researchers recorded the pRIR alongside the mean velocity of the last repetition.

## Training

Strength training sessions were conducted using free-weight Eleiko equipment.

As a general warm-up, five minutes of stationary cycling at 80–120 W (Keiser M3i Lite, Keiser Sport, Fresno, CA, USA) was followed by 5–10 min of dynamic shoulder mobility. Finally, 3–5 specific warm-up sets were conducted, progressing from ~40% of 1RM (5–10 repetitions).

The individual training loads for each training session were determined by pre-planned target velocities (Table 2). Before the training sets, a load-test was conducted. Loads (kg) were adjusted to achieve the pre-planned mean velocity with a maximal deviation of 0.03 ms$^{-1}$. This was typically achieved by 1–3 single repetition sets with maximal effort in the ascending phase. To exemplify, if the target velocity was 0.53 ms$^{-1}$, a velocity of 0.50–0.56 ms$^{-1}$ was acceptable. Based on pre-set velocities, the velocity thresholds for set termination were determined (these thresholds were, thus, different for the LVL and HVL groups; Table 2). Both groups completed three training sessions, featuring "low," "moderate," and "heavy" loads (Table 2).

Both the bench press and the squat exercise were executed in accordance with the rules of powerlifting (http://www.powerlifting.sport), and stance width, grip width, and range of motion were individually standardized across the 18 training sessions. All sessions were supervised to ensure proper lifting techniques, and real-time feedback was given to participants. Sets were terminated if improper technique was observed. An experienced spotter was present to ensure lifter safety but intervened only in cases of unexpected failure or perceived risk of injury. Interrupted sets were rare and excluded from the analysis.

## Statistics

All analyses and presentations are done with pRIRs from 0 to 4 to ensure sufficient statistical power at each pRIR value. 145 measurements with pRIR >4 were excluded (4.7% of the total data points). The excluded pRIR data points ranged from 5 to 12, and when divided by group, session (load), and sex, the number of data points per pRIR (5–12) was very low.

Pearson's r was used to calculate correlations between variables. The association between mean velocity and pRIR was calculated for each participant and then averaged for grouped values.

Effect sizes (ES) were estimated using Cohen's d, and the following scale was used: <0.2 trivial, 0.2–0.6 small, 0.6–1.2 moderate, 1.2–2.0 large, 2.0–4.0 very large. For correlations, the following scale was used: 0.1–0.3 small, 0.3–0.5 moderate, 0.5–0.7 large, 0.7–0.9 very large (Hopkins et al., 2009).

We fit a linear mixed-effects model (LMM) to assess the relationship between mean velocity and pRIR, accounting for repeated measurements and individual variability. pRIR was treated as a continuous, approximately normal outcome. This approach was chosen for its interpretability, as it allows direct estimation of fixed effects on the original scale. Sensitivity analyses were performed using alternative model approaches. These models yielded similar fixed-effect estimates and did not change the overall conclusions, supporting the use of the linear model.

The model included fixed effects for exercise type (bench press, squat), velocity group (LVL, HVL), load, sets per session, sex (female, male), and training week. Additionally, interaction terms between exercise type and velocity group were included to evaluate whether the relationship between velocity and pRIR varied by exercise type. To capture between-subject variability, the model incorporated random intercepts and random slopes for each participant. The random intercepts accounted for individual differences in baseline pRIR, while random slopes for mean velocity allowed individuals to vary in their response to velocity. The correlation between random intercepts and slopes was estimated to assess whether participants with higher baseline pRIR responded differently to changes in velocity. Residual variance accounted for within-subject fluctuations, representing variation in pRIR that the fixed or random effects did not explain. The model was estimated using restricted maximum likelihood (REML), ensuring appropriate weighting of individuals contributing varying numbers of data points while preventing bias. Model estimates were reported with 95% confidence intervals, and statistical significance was set at $p < 0.05$.

The data were organized and analyzed using Python (3.11) with the pandas library (2.2.2). Plots and visualizations were created using matplotlib (3.9.2) and seaborn (version 0.13.2). The LMM was implemented in R (4.4.2) using the lme4 (1.1-37) package.

## RESULTS

A total of 2,972 mean velocity and pRIR combinations were analyzed. Table 2 presents a summary of the number of lifts differentiated by exercise, velocity loss group (LVL and HVL), sex (male and female), and type of session (low, moderate, or high loads).

The individual regression lines in Fig. 1 depict the relationship between mean velocity and pRIR (measured per set) for each participant. For the squat exercise, the correlation coefficients (r-values) between mean velocity and pRIR ranged from 0.3 to 0.9, with an average correlation of $0.6 \pm 0.2$ (mean $\pm$ standard deviation) across participants. For the bench press, the individual correlations ranged from 0.1 to 0.9, with an average correlation of $0.6 \pm 0.2$.

In Figs. 2–8, pRIR is plotted as a function of mean velocity. The data is differentiated on exercise, velocity group (LVL vs. HVL), type of session (low, moderate, and high load), and sex. Table 3 gives an overview of the number of data points for the different plots.

The LMM analysis (Table 4) revealed that the mean velocity was strongly associated with pRIR ($\beta = 6.99$, $p < 0.001$), indicating that faster movements correspond with higher pRIR values. Group (LVL vs. HVL) was the second strongest predictor ($\beta = 0.94$, $p < 0.001$), suggesting that individuals in the HVL group tend to report higher pRIR. Exercise type demonstrated a negative effect ($\beta = -0.67$, $p < 0.001$), meaning that the bench press was associated with lower pRIR values than the squat. The interaction between group and exercise type was significant and negative ($\beta = -0.33$, $p < 0.001$), which suggests that the difference in pRIR between squat and bench press was smaller in the HVL group than in the LVL group. Additionally, the number of sets per session was inversely related to pRIR ($\beta = -0.11$, $p < 0.001$). Weeks of training ($\beta = -0.01$, $p = 0.27$) and sex ($\beta = 0.35$, $p = 0.26$) did not significantly affect pRIR. The random effects revealed considerable between-subject

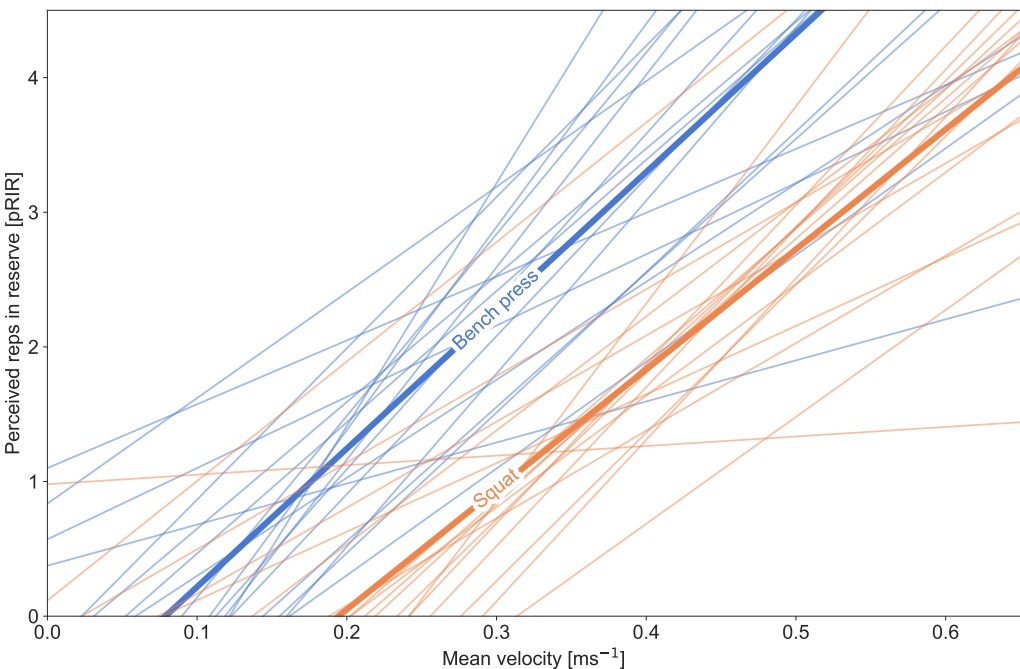

**Figure 1 Individual regression lines for the correlation between average concentric velocity (ms⁻¹) for each reported value of perceived repetitions in reserve (pRIR).** The blue lines represent the bench press, and the orange represents the squat. The thick lines represent the mean of the individual lines for each exercise.

variability, with a standard deviation of 0.96 for the intercept and 2.25 for the slope of mean velocity and a strong negative correlation ($r = -0.73$) between these random components. The within-subject residual variability was 0.69.

## DISCUSSION

This study explored the relationship between mean velocity of the bar and pRIR across a 6-week training period, analyzing nearly 3,000 data points. The key findings were: (1) The mean velocity was strongly associated with pRIR but with individual differences from trivial to very large correlations. (2) The type of exercise, *i.e.*, bench press and squat, the pre-set velocity loss thresholds (LVL or HVL), the session load (% of 1RM), and the number of sets influenced the relationships between bar velocity and pRIR. (3) The pRIR was not appreciably influenced by sex and the number of training weeks.

Perceived or predicted RIR (pRIR) is a subjective variable with inherent biases (*Hackett et al., 2017*; *Hackett et al., 2012*; *Helms et al., 2017a*). *Hackett et al. (2012)* observed that the participants tended to underestimate actual RIRs in the first sets (sets 1 and 2) but more accurately predicted RIR in the later sets (sets 3 and 4). *Hackett et al. (2017)* reported an accuracy of ~1 repetition when the actual RIR was 0–5, while the inaccuracy increased to >2 repetitions with an actual RIR of 7–10. These observations were supported by *Zourdos et al. (2021)*. In our study, pRIRs were <5, suggesting that participants likely demonstrated

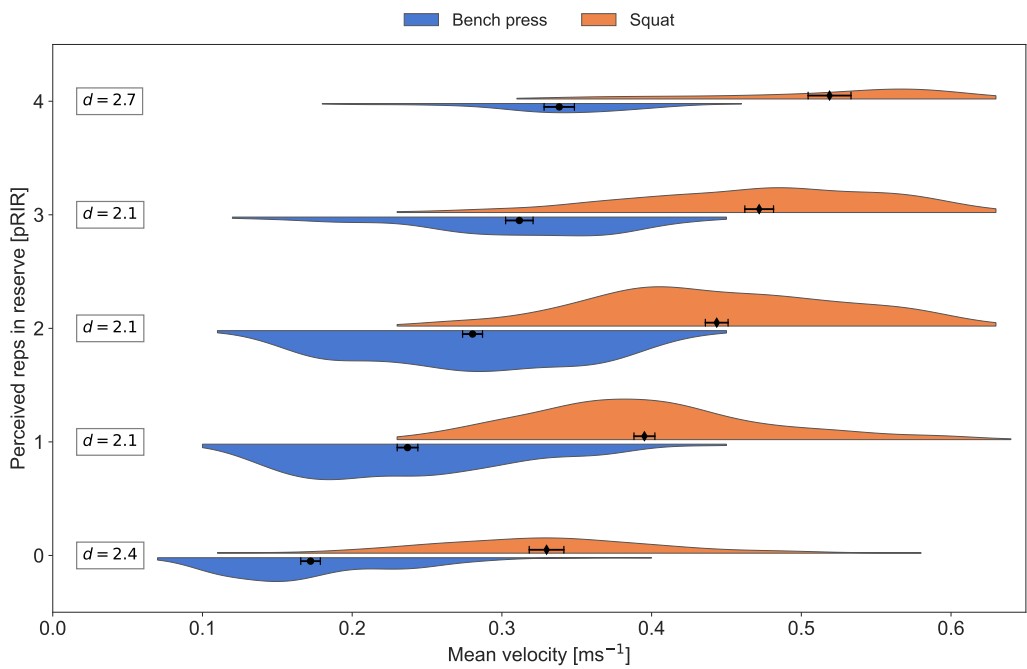

**Figure 2** **Average concentric velocity and perceived repetitions in reserve for the bench press and the squat.** Kernel density estimation plots of average concentric velocity (ms$^{-1}$) for each reported value of perceived repetitions in reserve (pRIR). Comparison of exercise; bench press (blue) *vs.* squat (orange). Error bars are 95% confidence intervals around the mean. d = Cohen's d, effect size for the difference.

acceptable RIR estimation accuracy. In support of this, the LMM analysis showed a within-subject residual variability of 0.69, indicating that the participants' reported pRIR values, on average, deviated from the model's predictions by less than one repetition.

All participants reported lower pRIR for a given mean velocity in the squat than in the bench press. This aligns with observations that 1RM is achieved with a significantly lower mean velocity in the bench press (~0.1–0.2 ms$^{-1}$) than in the squat (~0.2–0.3 ms$^{-1}$) (*Gonzalez-Badillo & Sanchez-Medina, 2010*; *Helms et al., 2017b*; *Iglesias-Soler et al., 2019*; *Zourdos et al., 2016*). The greater range of motion and bar displacement, combined with a shorter sticking region and stronger acceleration in the post-sticking phase of the squat compared to the bench press, may explain the difference (*Kompf & Arandjelović, 2017*; *van den Tillaar, Andersen & Saeterbakken, 2014*).

In the present study, about half of the participants trained with low bar velocity loss thresholds (LVL) for terminating each set (*i.e.,* 20% for squat and 30% for bench press), while the other half trained with high bar velocity loss thresholds (HVL: 40% in the squat and 60% in the bench press)—*i.e.,* close to contraction failure in each set. Visual inspection of Figs. 3 and 4 shows that the HVL group reported higher pRIR values for a given mean velocity than the LVL group, which was most pronounced for the bench press exercise. This finding was confirmed by the LMM, which detected a significant group effect of about one pRIR and an interaction effect between exercise type and velocity group. By design, the LVL group terminated the sets at higher mean velocities than the HVL

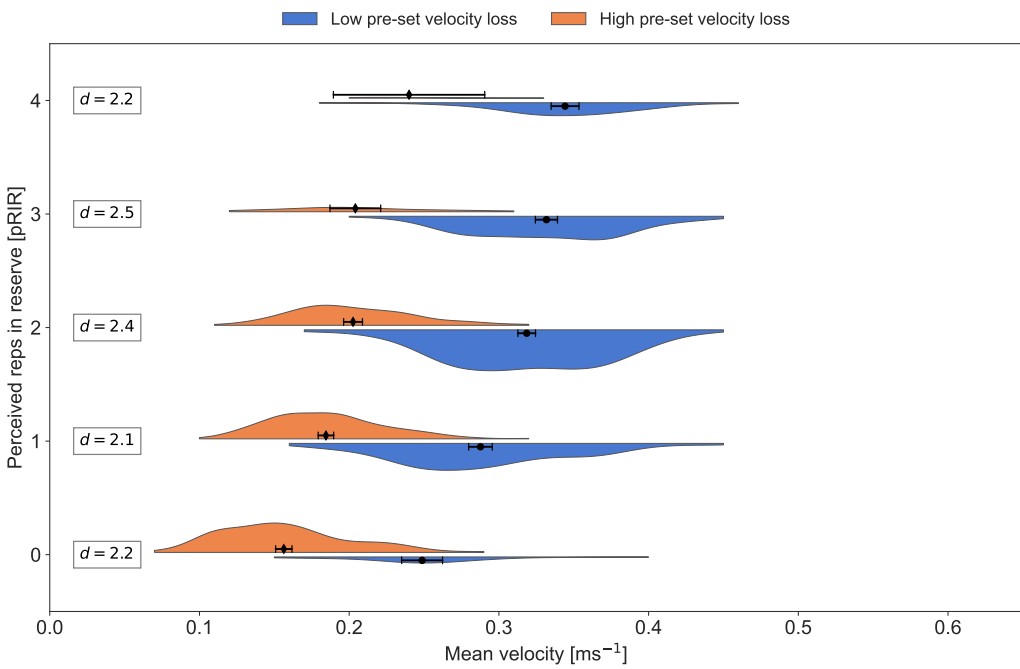

**Figure 3** **Average concentric velocity and perceived repetitions in reserve in the bench press for the low- and high-velocity-loss group.** Kernel density estimation plots of average concentric velocity (ms$^{-1}$) for each reported value of perceived repetitions in reserve (pRIR) in bench press. Comparison of pre-set velocity loss threshold group; low-velocity-loss group (blue) *vs.* high-velocity-loss group (orange). Error bars are 95% confidence intervals around the mean. d = Cohen's d, effect size for the difference.

group, meaning the LVL group was likely to underestimate the RIR. This suggestion aligns with observations from previous studies (*Hackett et al., 2012*; *Zourdos et al., 2021*), which have shown systematically more accurate RIR prediction closer to failure. Nonetheless, it remains possible that the HVL group overestimated their RIR.

Our participants conducted three different sessions per week, consisting of low-, medium-, and high-load sessions (Table 2). We observed a clear load effect on pRIR, with low and moderate loads resulting in lower pRIR values at a given mean velocity compared to heavy loads. This was evident for both exercises, especially for the squat (Figs. 7 and 8). Interestingly, the sensation and experience of effort and exertion seem to depend on several factors, including neural drive from the motor cortex, which increases with higher muscle force requirements, and neuromuscular fatigue (*Pageaux, 2016*). Heavy loads challenge the lifter's maximal capacity, where even minimal fatigue can lead to failure. In contrast, lighter loads allow for continued repetitions despite severe neuromuscular fatigue (*Behm et al., 2002*). Thus, the sensation of muscle fatigue and discomfort (*Pollak et al., 2014*) might lead to an underestimation of RIR with lower loads. Alternatively, the participants were overly optimistic in their RIR predictions with heavy loads due to low muscular fatigue and discomfort, "it just feels heavy". In correspondence with this, *Mansfield et al. (2023)* observed an effect of load (60% *vs.* 80% of 1RM) in the prediction (estimates) of RIR in the bench press, where the mean velocity at 0–3 RIR was higher with 60% of 1RM load
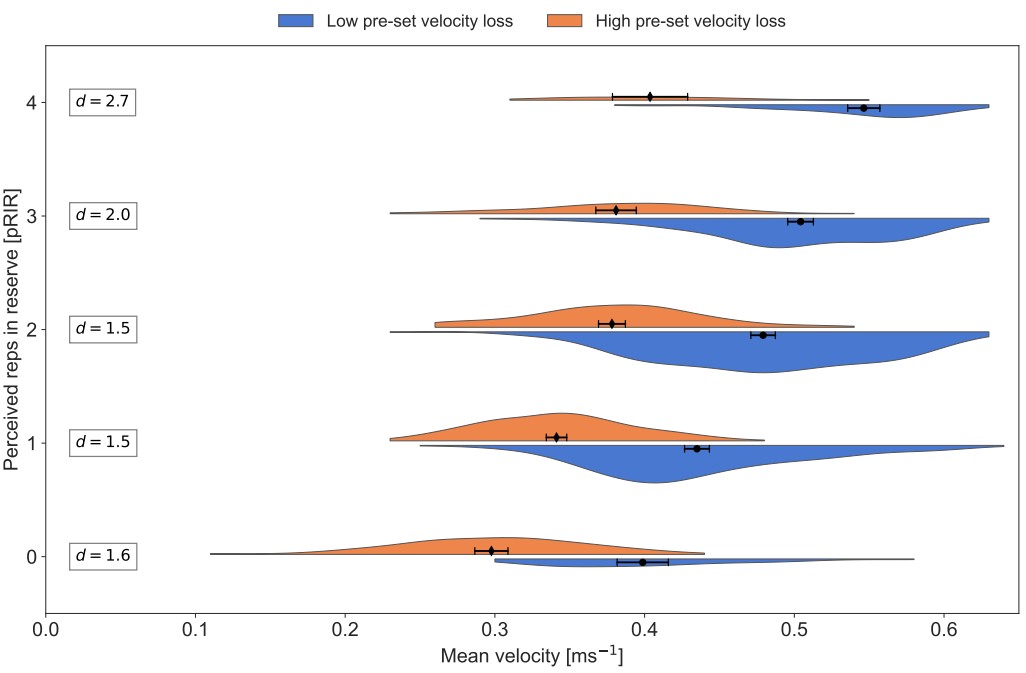

**Figure 4  Average concentric velocity and perceived repetitions in reserve in the squat for the low- and high-velocity-loss group.** Kernel density estimation plots of average concentric velocity (ms$^{-1}$) for each reported value of perceived repetitions in reserve (pRIR) in squat. Comparison of pre-set velocity loss threshold group; low-velocity-loss group (blue) *vs.* high-velocity-loss group (orange). Error bars are 95% confidence intervals around the mean. d = Cohen's d, effect size for the difference.

than with 80% of 1RM. Physiologically, this suggests that neuromuscular fatigue during low loads—likely due to an increased number of repetitions—causes a steeper decline in mean velocity during the final repetitions of a set compared to heavier loads with fewer total repetitions performed (hence, the drop in velocity may be non-linear).

*Mansfield et al. (2023)* observed that RIR decreased with each successive set. We found the same; for a given mean velocity, the pRIR was lower in, for example, the third set compared to the first. This can be due to neuromuscular fatigue per se and/or the fact that the participants became less optimistic about their RIRs for a given mean velocity due to the perception of fatigue. This effect was, nevertheless, rather small, accounting for about −0.1 RIR per set (according to the LMM analysis).

We observed that the correlations between mean velocity and pRIR varied considerably at the individual level, but no systematic improvements (changes) in the relationship between the mean velocity and the pRIR occurred during the 6-week training period, inferring that the participants did not improve (or worsen) their RIR prediction accuracy. This finding is in agreement with recent studies (*Halperin et al., 2022; Jukic et al., 2024; Remmert, Laurson & Zourdos, 2023*) that revealed no apparent effects of training status (experience) on the accuracy of RIR prediction. However, it would be intriguing to see if individuals with poor correlations between bar velocity and pRIR could improve by

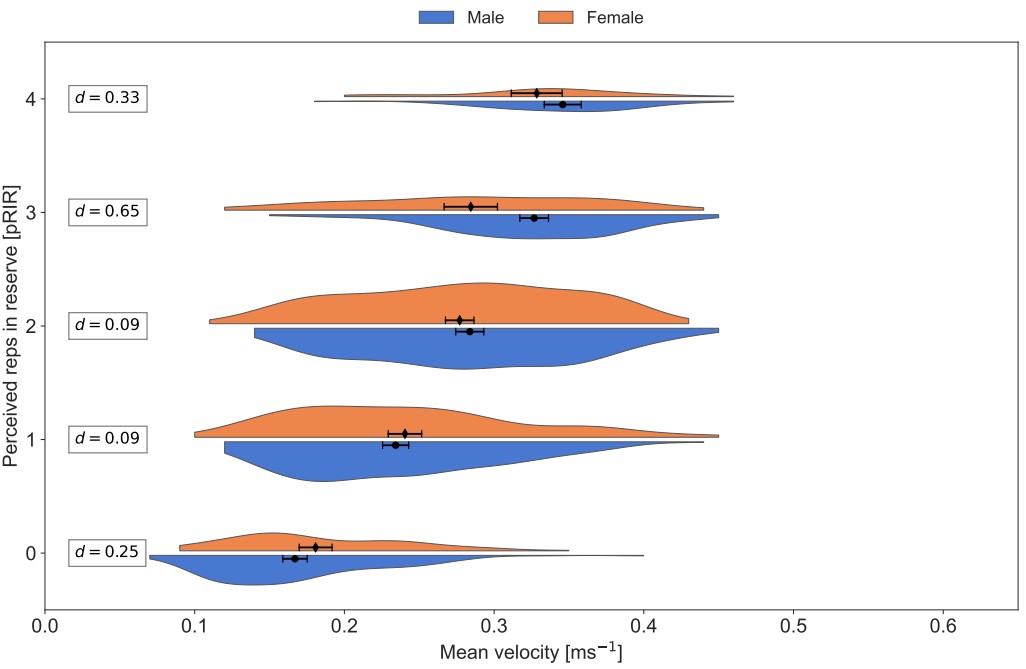

**Figure 5** **Average concentric velocity and perceived repetitions in reserve for the bench press in males and females.** Kernel density estimation plots of average concentric velocity (ms$^{-1}$) for each reported value of perceived repetitions in reserve (pRIR) in bench press. Comparison of sex; males (blue) *vs.* females (orange). Error bars are 95% confidence intervals around the mean. d = Cohen's d, effect size for the difference.

deliberate practice, as no feedback was given during the training period in the current study.

Subjective assessments of effort and exertion (such as RPE and RIR) are reported to be practically similar between sexes across different modes of exercise and exertion (*Losnegard et al., 2021*; *Morishita et al., 2018*; *Naclerio & Larumbe-Zabala, 2017a*; *Naclerio & Larumbe-Zabala, 2017b*). This is in line with our observations; the effect of sex was not significant in the LMM analysis. From visual inspection of Figs. 5 and 6, we can, however, see a trend where the females recorded higher pRIR values at a given bar velocity. Accordingly, *Odgers et al. (2021)* observed lower bar velocities in females than males at 7-9 RPE (corresponding to 1-3 RIR) in the front squat exercise but not in the hexagonal bar deadlift exercise. Moreover, *Naclerio & Larumbe-Zabala (2017b)* observed no sex difference in squats in strength-trained athletes. Generally, females demonstrate similar or better neuromuscular endurance than males, which might, at least partly, be related to absolute lower strength and an on average higher distribution of type I fiber in females than males (*Hunter, 2009*). Note, however, that we had a limited number of participants (both females and males), causing a risk of missing a potential sex effect.

A key limitation of this study was measuring only pRIR, not actual RIR, precluding quantification of true estimation precision. To mitigate this, our analysis was restricted to a pRIR of 0–4, as estimations within this range are generally more accurate and considered

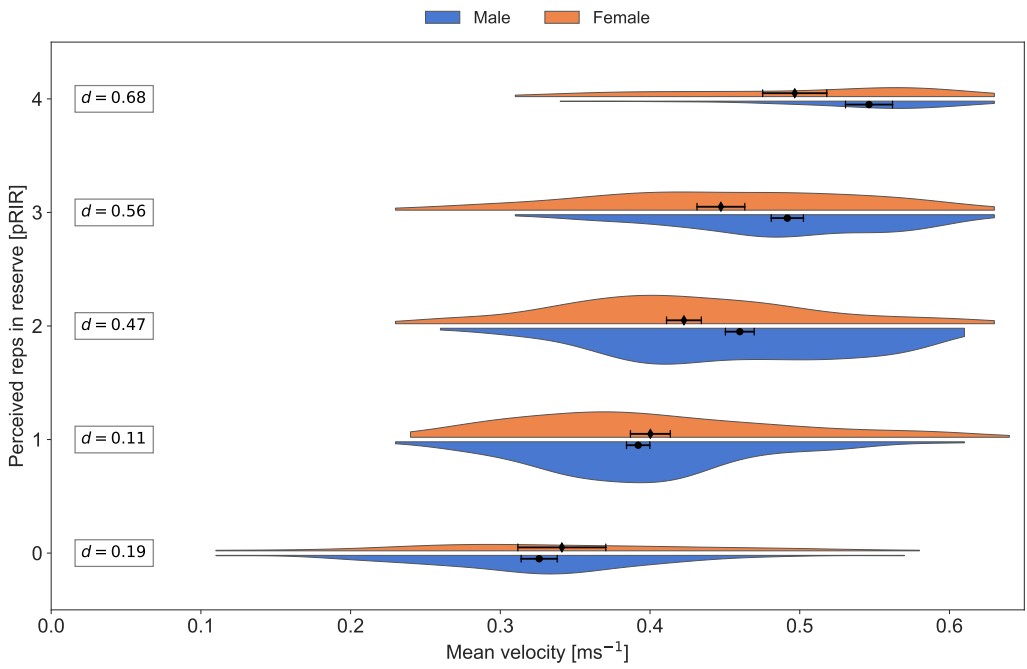

**Figure 6** **Average concentric velocity and perceived repetitions in reserve for the squat in males and females.** Kernel density estimation plots of average concentric velocity (ms$^{-1}$) for each reported value of perceived repetitions in reserve (pRIR) in squat. Comparison of sex; males (blue) *vs.* females (orange). Error bars are 95% confidence intervals around the mean. d = Cohen's d, effect size for the difference.

practically acceptable (*Bastos, Machado & Teixeira, 2024*; *Halperin et al., 2022*; *Jukic et al., 2024*).

Another potential limitation is the use of mean velocity. It could be that mean propulsion velocity (the mean velocity of the acceleration phase) and/or the lowest/minimum velocity had resulted in different results and should be further investigated; however, with the loads used in the present study (~77–89% of 1RM), the differences between mean propulsion and mean velocities of the full movement should be very small or negligible (*Sanchez-Medina et al., 2017*; *Sanchez-Medina, Perez & Gonzalez-Badillo, 2010*).

Despite its limitations and descriptive nature, the present study has high ecological value by virtue of an extensive data set from actual training sessions conducted by well-strength-trained individuals. Moreover, all sessions were supervised. The inclusion of both sexes and different pre-set velocity loss targets (low and high), as well as sessions with different loads (low, medium, and heavy) in both bench press and squat (an upper and a lower body exercise), provided us with several interesting observations that allude to important nuances to be aware of when applying VBST and pRIR in training.

## PRACTICAL IMPLICATIONS

Different strength exercises, *e.g.*, bench press and squat, exhibit distinct velocity-RIR profiles at the individual level. Mean velocity and pRIR are complementary but not interchangeable

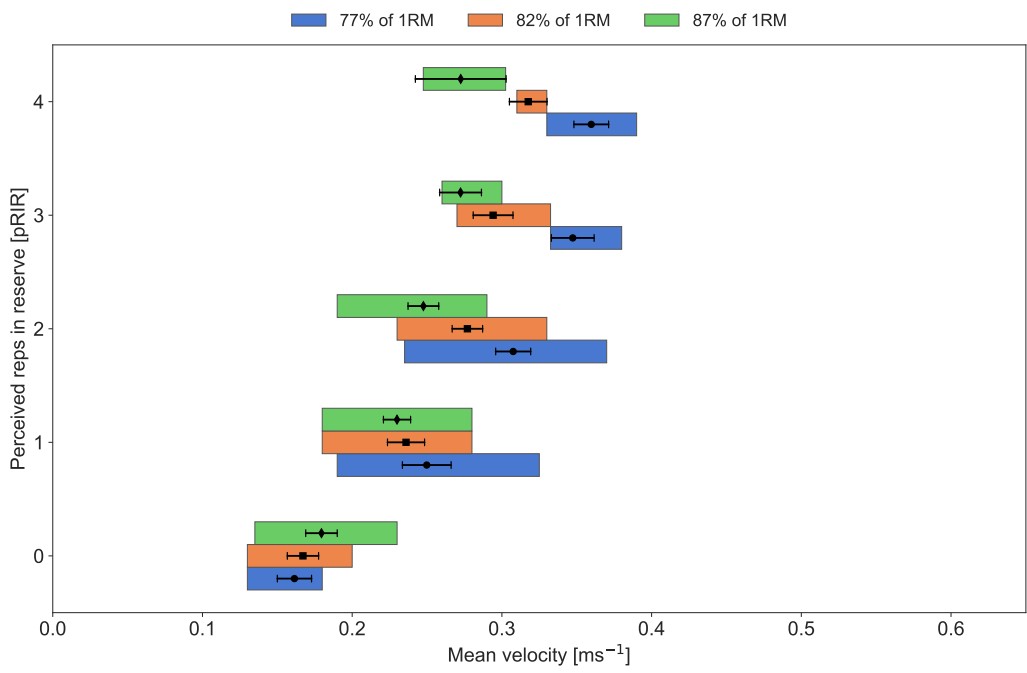

**Figure 7** **Average concentric velocity and perceived repetitions in reserve in the bench press with different training loads.** Box plots showing median, lower, and upper quartiles of average concentric velocity (ms$^{-1}$) for each reported value of perceived repetitions in reserve (pRIR) in **bench press**. Comparison of loads in % of 1 repetition maximum; low (77% = 0.53 ms$^{-1}$, blue), moderate (82% = 0.45 ms$^{-1}$, orange), and high (87% = 0.38 ms$^{-1}$, green). Error bars are 95% confidence intervals around the mean.

methods, as also noted by *Mansfield et al. (2023)*. Consequently, training sessions managed by VBST or pRIR may differ, even when the intended session goal is the same. However, we do not assert that one method is superior to the other, as both have advantages and limitations. For power development and maximal strength training, VBST offers a key advantage over pRIR, as higher RIR values can compromise estimation accuracy, and bar velocity is directly relevant for assessing performance and training quality *via* biofeedback (*Weakley et al., 2021*). In contrast, pRIR may be equally effective or even preferable for hypertrophy-focused training, particularly when working within 0–2 RIR (*Grgic et al., 2022*; *Halperin et al., 2022*; *Suchomel et al., 2021*). Combining both methods may provide optimal benefits for high-level athletes, whereas RIR alone is likely sufficient for beginners, recreational lifters, and non-athlete populations, such as patients (*Bastos, Machado & Teixeira, 2024*; *Maroto-Izquierdo et al., 2025*). Further research across diverse populations is needed to better evaluate the advantages of VBST compared to or in combination with pRIR, given that pRIR is cost-free and easy to implement.

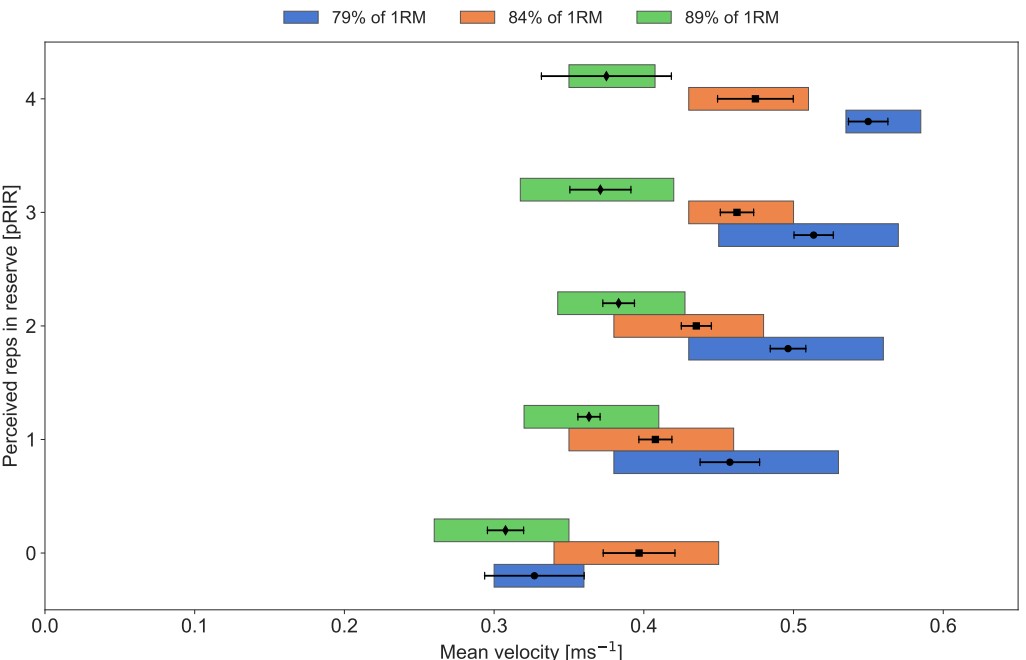

**Figure 8 Average concentric velocity and perceived repetitions in reserve in the squat with different training loads.** Box plots showing median, lower, and upper quartiles of average concentric velocity (ms$^{-1}$) for each reported value of perceived repetitions in reserve (pRIR) in squat. Comparison of loads in % of 1 repetition maximum; low (79% = 0.70 ms$^{-1}$, blue), moderate (84% = 0.6 ms$^{-1}$, orange) and high (89% = 0.49 ms$^{-1}$, green). Error bars are 95% confidence intervals around the mean.

**Table 3 Overview of correlation analyses presented in Figs. 2–8.** The x-axis is either absolute velocity (ms$^{-1}$) in the final repetition of the set or the percentage (%) loss in velocity. The data are filtered for exercise, sex, group (LVL, HVL), and session (low, moderate, or high loads).

| Figure | Exercise | Filter | X-axis | Y-axis | N |
|---|---|---|---|---|---|
| 2 | Both | Exercise; bench press *vs.* squat | Mean velocity (ms$^{-1}$) | RIR; 0-4 | 1,476 *vs.* 1,496 |
| 3 | Bench Press | Velocity group; LVL *vs.* HVL | Mean velocity (ms$^{-1}$) | RIR; 0-4 | 853 *vs.* 623 |
| 4 | Squat | Velocity group; LVL *vs.* HVL | Mean velocity (ms$^{-1}$) | RIR; 0-4 | 921 *vs.* 575 |
| 5 | Bench Press | Sex; female *vs.* male | Mean velocity (ms$^{-1}$) | RIR; 0-4 | 835 *vs.* 641 |
| 6 | Squat | Sex; female *vs.* male | Mean velocity (ms$^{-1}$) | RIR; 0-4 | 881 *vs.* 615 |
| 7 | Bench Press | Session loads; low (0.53 ms$^{-1}$) *vs.* moderate (0.45 ms$^{-1}$) *vs.* high (0.38 ms$^{-1}$) | Mean velocity (ms$^{-1}$) | RIR; 0-4 | 490 *vs.* 484 *vs.* 502 |
| 8 | Squat | Session loads; low (0.70 ms$^{-1}$) *vs.* moderate (0.60 ms$^{-1}$) *vs.* high (0.49 ms$^{-1}$) | Mean velocity (ms$^{-1}$) | RIR; 0-4 | 465 *vs.* 512 *vs.* 519 |

**Notes.**

$N$, number of data points; HVL, high-velocity-loss threshold group; LVL, low-velocity-loss threshold group; RIR, repetitions in reserve.

**Table 4 Generalized linear mixed models with perceived repetitions in reserve (pRIR) as the dependent variable.**

| Predictor of pRIR | Effects | Coefficient ($\beta$) | Std. error | z-value | p-value | 95% Confidence interval |
|---|---|---|---|---|---|---|
| Intercept (*i.e.*, when mean velocity is 0) | Fixed | −0.59 | 0.27 | −2.2 | 0.04 | [−1.15, −0.03] |
| Mean velocity (ms$^{-1}$) | Fixed | 6.99 | 0.59 | 11.8 | 0.00 | [5.77, 8.21] |
| Group (LVL *vs.* HVL) | Fixed | 0.94 | 0.07 | 14.2 | 0.00 | [0.81, 1.07] |
| Exercise (squat *vs.* bench press) | Fixed | −0.67 | 0.07 | −9.8 | 0.00 | [−0.80, −0.53] |
| Sex (female *vs.* male) | Fixed | 0.35 | 0.30 | 1.2 | 0.26 | [−0.29, 0.99] |
| Load (low *vs.* high) | Fixed | 0.35 | 0.04 | 8.3 | 0.00 | [0.27, 0.43] |
| Interaction: LVL *vs.* HVL x squat *vs.* bench press | Fixed | −0.33 | 0.07 | −4.9 | 0.00 | [−0.46, −0.19] |
| Load (moderate *vs.* high) | Fixed | 0.23 | 0.03 | 6.8 | 0.00 | [0.17, 0.30] |
| Sets (number of sets for each exercise per session) | Fixed | −0.11 | 0.01 | −14.1 | 0.00 | [−0.12, −0.09] |
| Week (1–6 of the training period) | Fixed | −0.01 | 0.01 | −1.1 | 0.27 | [−0.02, 0.01] |
| SD of the intercept (mean velocity) | Random (between-subject) | 0.96 | | | | |
| Correlation between the intercept and mean velocity | Random (between-subject) | −0.73 | | | | |
| SD of the random slope of mean velocity | Random (between-subject) | 2.25 | | | | |
| SD of the residual variability | Random (within-subjects) | 0.69 | | | | |

**Notes.**
HVL, high-velocity-loss threshold group; LVL, low-velocity-loss threshold group; Load, session loads (squat: low = 0.70 ms$^{-1}$, moderate = 0.60 ms$^{-1}$, and high = 0.49 ms$^{-1}$; bench press, low = 0.53 ms$^{-1}$, moderate = 0.45 ms$^{-1}$, and high = 0.38 ms$^{-1}$); SD, Standard deviation.

## CONCLUSION

In this study, we investigated the relationship between mean velocity of the bar and pRIR during multiple exercise sessions. We conclude that type of exercise (bench press and squat), load (in %RM), the number of sets, and the proximity to failure (velocity loss threshold), but not sex and the number of training weeks, influenced the relationships between mean bar velocity (of the ascending phase) and pRIR. Both mean velocity and pRIR can be used to manage strength training, but these metrics may not be used interchangeably. We recommend using VBST in conjunction with pRIR for optimal control during strength training.

## ACKNOWLEDGEMENTS

The authors would like to thank the participants for their time and efforts.

### Funding

The authors received no funding for this work.

### Competing Interests

The authors declare there are no competing interests.

### Author Contributions

- Gøran Paulsen conceived and designed the experiments, performed the experiments, analyzed the data, prepared figures and/or tables, authored or reviewed drafts of the article, and approved the final draft.
- Roger Myrholt conceived and designed the experiments, performed the experiments, analyzed the data, prepared figures and/or tables, authored or reviewed drafts of the article, and approved the final draft.
- Fredrik Mentzoni analyzed the data, prepared figures and/or tables, authored or reviewed drafts of the article, and approved the final draft.
- Paul Andre Solberg conceived and designed the experiments, analyzed the data, authored or reviewed drafts of the article, and approved the final draft.

### Ethics

The following information was supplied relating to ethical approvals (*i.e.*, approving body and any reference numbers):

The study was approved by the local ethical committee at the Norwegian School of Sport Sciences (103-290819).

### Data Availability

The raw data is available in the Supplemental File.

### Supplemental Information

Supplemental information for this article can be found online at http://dx.doi.org/10.7717/peerj.19797#supplemental-information.

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
