# Peer review of "Exercise type, training load, velocity loss threshold, and sets affect the relationship between lifting velocity and perceived repetitions in reserve in strength-trained individuals"

_PeerJ, doi:10.7717/peerj.19797_

## Round 0.1 · original submission · Major Revisions

The manuscript has been well-received by the reviewers and provides novel insights to advance the field of research. However, there are some significant areas for improvement, particularly in the methods section. Please ensure the methods section appropriately describes the procedures so a reader can reproduce the experiment.

Please address reviewer comments relating to the clarity of writing or consistency in using terms and abbreviations. For example, you provide an acronym for velocity-based strength training (VBST) in line 40 but use the entire term in line 41.

Please provide a point-by-point response to the reviewer comments in your revision.

We look forward to reviewing your amended manuscript.

Reviewer 1 ·

Basic reporting

The authors have submitted the manuscript entitled ‘Exercise type, training load, velocity loss threshold, and sets affect the relationship between lifting velocity and perceived repetitions in reserve in strength-trained individuals’, which is understandable and well referenced. They address a very interesting issue for fitness coaches regardless of results, since the relationship between velocity loss and potential repetitions might have a link as obvious as unknown. As the authors have reported, many constraints might influence their observations, but the ecological approach provides a step beyond understanding the RIR and the velocity-loss relationship. However, the authors need to improve major weaknesses such as the strength and cohesion of the introduction section and reporting the experimental design, which must ensure the reproducibility of the study, and other minor weaknesses such as style references and missing citations. All of them are detailed in the following sections.

Experimental design

As I said in the previous form, the topic is quite interesting, but the method section must be rewritten, including proper subsections like procedures, instruments, and statistical analysis separately. Additionally, despite the manuscript being based on a study already published, the authors need to provide enough information to suggest what they did.

Validity of the findings

Awaiting further information.

Additional comments

Lines 24-25. I suggest being consistent with the literature and using the ‘Velocity Mean Propulsive’ instead of ‘Average Concentric Velocity’ since the bar, in which the velocity is measured, does not cause a contraction.
Lines 57-58. Please include this sentence in the text before so that the paragraph has cohesion.
Line 78. Cite these previous studies.
Lines 92-94. Clarify if subjects performed each load. Include in the study design section a specific paragraph on the kind of study and the chronological study flow chart.
Lines 113-114. How many subjects were assigned in each group? Was the sex mixed?
Line 114. I think the percentage of velocity loss was moved. A 30% lost velocity may be too close to muscle failure to be considered a 'Low-velocity loss threshold' in the bench press.
Line 115. The same concerns as before.
Lines 116-118. Cite the studies on which you have relied.
Line 125. Please specify what type of machine was used (Smith machine or free weight).
Line 126. The warming should be described or cited from the study that has been followed.
Line 134. Provide the number of series interrupted and clarify if they were included in the analysis.
Lines 141-142. Describe the placement and care of the encoder use. What were the criteria?
Line 147. Please merge this sentence with a paragraph in which the training is briefly explained.
Line 151. Was the last repetition recorded by the encoder only? Was there any measurement to achieve the maximum repetitions in each load? (Just to ensure that the subject has a reference and pRIR might be slightly accurate). Include it.
Lines 158 & 160. Provide a reason for excluding the observation from the analysis. I am afraid of statistical power, which is not a good reason, since as far as observations are concerned, the more the better.
Line 162. Please remove any authors’ comments from the manuscript.
Line 164. Could you please indicate which family was set up in the model? I assume that binomial was used.
Line 175. You need to show the results by indicating how you interpret them. In this way, you increase the understandability of the results section.
Line 183. What do you mean? Please, provide an identification of each variable analysed in each correlation.
Line 219. You might wish to discuss the influence of the type of muscle (Fibre arrangement) and joint involved in each exercise. Please note that pull-up has a similar range of movement to squat, but the mean velocity is closer to bench press.
Line 221. I have the main concerns about this. According to this paragraph, the subjects only trained in one level of velocity loss, but for running a generalised linear mixed model you need to identify each subject in the model. Therefore, I suspect that the regression was performed by merging different observations among subjects. Provide more information since the model might have been carried out inappropriately. Note that the slope in the power-force-speed profile has a large individual component that must be taken into account.
Line 225. The literature shows that differences between 1RM velocities may not be observed between close percentages, so individual variability may be masked by this feature. Standardisation of the values obtained in each measurement (e.g., relative 1RM or slope of the force-velocity curve for each subject) would be necessary. What measurement was considered in the analysis of this?
Line 274. Yes, that is a big weakness, but you might not be able to do everything you want in an ecological design. Nevertheless, you have a lot of information to provide other evidence, like exploring what variables are suitable for predicting pRIR (i.e., you might perform a backward regression).
Line 301. Citation form: Author (year)
Line 307 and the following lines. Please check the referencing style; in some cases, there is a lack or excess of information (Url is not DOI).

Reviewer 2 ·

Basic reporting

The manuscript is written in professional English. However, minor improvements can be made to streamline sentences for better clarity. References are current and sufficiently support the background and rationale. Briefly discussing any contrasting views from recent meta-analyses can enhance the discussion. Raw data is available, complying with best practices.

Experimental design

The manuscript is written in professional English. However, minor improvements can be made to streamline sentences for better clarity. References are current and sufficiently support the background and rationale. Briefly discussing any contrasting views from recent meta-analyses can enhance the discussion. Raw data is available, complying with best practices.

Validity of the findings

Statistical analyses are appropriate for the study's design.
While the discussion effectively contextualizes findings, additional implications for broader populations (e.g., untrained individuals) could be explored in future studies.

Additional comments

The manuscript addresses a relevant topic with a solid methodological foundation and presents significant findings. The use of a large dataset is a considerable strength. However, the reliance on subjective perceived repetitions in reserve without direct validation against actual RIR introduces potential bias. The manuscript is well-written, but minor grammatical corrections are needed, please, revise the entire manuscript
Specific comments:
- Please, consider rephrasing the conclusion in the abstract, including a clear answer to the purpose of the study.
- Please, consider explicitly stating in the introduction how the findings may influence strength training prescriptions beyond the observed conditions (e.g., other populations or untrained individuals).
- The purpose presented at the end of the introduction is not the same as in the abstract. Please, clarify. Also, the last sentence of the introduction seems to be part of the methodology.
- Was there any inclusion and/or exclusion criteria for the participants?
- The stratification of participants into low- and high-velocity-loss groups is well-justified but could benefit from a more detailed rationale for the threshold values chosen
- In the study design, the authors stated that “The training volume (total number of reps) between groups was matched.” Please, clarify.
- The authors state that the ACV (m/s) of the last repetition in each set was recorded and matched with the corresponding pRIR value. This means that the velocity of the last repetition was used for analysis. Why did the authors not use the velocity drop, the number of repetitions performed, but this value of velocity? Considering that the initial velocity should be the same (same %of 1RM), the velocity loss is the same within group (e.g., 20%), the last ACV should be the same in the participants of the same group. Please, explain the variables chosen. This is a major issue of manuscript design.
- Tables are informative but should include explicit units for all numerical values
- Please, consider including individual plots in Figure 1.

---

## Round 0.2 · Minor Revisions

There are areas for improvement in the clarity and conciseness of your research. Please refer to the specific comments provided by the reviewers and address or rebut these as necessary.

·

Basic reporting

I believe this paper is well written. Some rephrasing should take place to better convey the intended message. I believe the paper has an extensive number of figures that can be reduced. Overall, it is well structured.

Experimental design

The research question is clearly presented and its meaningfull. It fills a gap in the literature and it falls within the Aims and Scope of the journal. I mad some comments to enhance the manuscript's quality in a document I attach to this revision.

Validity of the findings

The findings are interesting from a merely descriptive standpoint. As the actual RIRs were not assessed, a lot of the conclusions are speculative. Regardless, this paper provides interesting insights on the influence of meaningful factors on the perception of RIR, and that is notworthy.

Additional comments

I commend the authors for this interesting paper. A document is attached to this revision with some cocnerns I found will revising your manuscript.

---

## Round 0.3 · accepted · Accept

Thank you for addressing the reviewer comments. You have satisfied the required amendments to warrant publication.

·

Basic reporting

Nothing to add.

Experimental design

Nothing to add.

Validity of the findings

Nothing to add.

Additional comments

Nice work with the revisions!